

# Enhanced soluble sugar content in tomato fruit using CRISPR/Cas9-mediated *SlINVINH1* and *SlVPE5* gene editing

Baike Wang[1], Ning Li[1,2], Shaoyong Huang[2], Jiahui Hu[1,3], Qiang Wang[1,2], Yaping Tang[1], Tao Yang[1], Patiguli Asmutola[1], Juan Wang[1] and Qinghui Yu[1]

[1] Institute of Horticulture Crops, Xinjiang Academy of Agricultural Sciences, Urumqi, China
[2] College of Forestry and Horticulture, Xinjiang Agricultural University, Urumqi, China
[3] Xinjiang Key Laboratory of Biological Resources and Genetic Engineering, College of Life Science and Technology, Xinjiang University, Urumqi, China

## ABSTRACT

Soluble sugar is known to improve the sweetness and increase tomato sauce yield. Studies have focused on improving the content of soluble sugar in tomato fruits, usually by promoting functional genes. We studied two genes (*SlINVINH1* and *SlVPE5*) that inhibited the accumulation of soluble sugar in tomato fruits and obtained two genes' knocked-out lines (CRISPR-*invinh1* or CRISPR-*vpe5*) using CRISPR/Cas9. Aggregated lines with CRISPR-*invinh1* and CRISPR-*vpe5* were gained by hybridization and self-pollination. Compared to wild-type lines, the glucose, fructose, and total soluble solid (TSS) contents of CRISPR-*invinh1* and CRISPR-*vpe5* increased significantly. Glucose, fructose, and TSS levels further improved simultaneously with CRISPR-*invinh1* and CRISPR-*vpe5* than with single gene knock-out lines. This indicates that these genes have a synergistic effect and will increase the soluble sugar content. Thus, the knock-out *SlINVINH1* and *SlVPE5* may provide a practical basis for improving the sweetness of tomato fruits and their processing quality.

# INTRODUCTION

Tomato (*Solanum lycopersicum*) is a popular and important commercial vegetable crop worldwide. However, modern cultivating practices have been criticized for producing unsatisfactory fruit flavors. Sweetness, an important taste characteristic of tomato fruits, is mainly determined by their soluble sugar content (*Mathieu et al., 2009*; *Tieman et al., 2017*). Soluble sugars influence fruit sweetness. They are also the most important component of the total soluble solids (TSS), which affects the production costs of processed tomatoes and tomato sauce. Studies have reported that an increase in processed tomatoes' TSS content from 4% to 5% could reduce raw material consumption by up to 25% when producing tomato sauce at a concentration of 28% (*Gur & Zamir, 2015*). Therefore, increasing the soluble sugar content in tomatoes improves fresh tomatoes' sweetness and increases the efficiency of tomato sauce production.

Corresponding authors
Juan Wang, 278089648@qq.com
Qinghui Yu, yuqinghui98@sina.com

Several factors directly influence the content of soluble sugars in tomato fruits, including genetic and environmental factors (including temperature, light, moisture, air, fertilizer, plant hormones). Technical factors such as cultivation and management play an indirect role through environmental factors (*Beckles, 2012*). However, genetic factors are fundamental and the most important. Although some studies have investigated genes related to soluble sugars, the corresponding biological functions of most of these genes are not fully understood (*Lupi et al., 2019*). This crop's immense economic value makes it necessary to conduct in-depth research and exploration of these genes. It is also necessary to combine this information with molecular marker-assisted selection or genetic engineering techniques to increase the soluble sugar content in cultivated tomatoes (*Gur & Zamir, 2015*; *Vallarino et al., 2017*). Understanding related genes aids the analysis of the molecular mechanisms for the accumulation of soluble sugars in tomato. This knowledge can also help us better understand other fruit types, which has substantial theoretical significance.

Plants have a set of complex metabolic mechanisms for sugars, including the sensing of sugar signals, sugar synthesis, loading, transportation, unloading, transformation, storage, accumulation, and the regulation of several physiological and biochemical processes. Several studies have focused on enzymes and proteins related to the processes of sugar synthesis, transportation, and decomposition. They include invertase, sucrose synthase (SUS), sucrose phosphate synthase (SPS), hexose kinases (Hexokinase, HXK), fructokinases (FRKs), sucrose phosphatase (Sucrose Phosphate Phosphatase, SPP), sugar transporters, and some transcription factors (*Beckles et al., 2012*; *Patrick, Botha & Birch, 2013*). These studies have facilitated a deeper understanding of the molecular mechanisms of sugar accumulation.

Invertase has a significant effect on sugar accumulation. Beets (*Beta vulgaris*) typically accumulate sucrose, while tomatoes largely accumulate hexose. The activity of insoluble acid invertase, also called cell wall invertase, decreases significantly during the early stages of sugar accumulation in beets. However, in tomatoes acid invertase activity increased significantly at the same stage (*Patrick, Botha & Birch, 2013*). A cloned QTL (Brix9–2–5) is involved the accumulation of soluble sugar is the *Lin5* gene. This gene is a cell wall invertase gene and is the key enzyme affecting the unload of sugar in tomato fruits. The soluble sugar content in the introgression line (IL) containing this site is significantly higher than in the IL without this site (*Baxter et al., 2005*). Silencing *Lin5* reduced the concentration of soluble sugar in the fruit (*Zanor et al., 2009*). Sugar accumulation can also be regulated by controlling the metabolic flow. Sucrose accumulation in sugarcane can generate feedback inhibition. When sucrose is converted to fructose and glucose is promoted by transgenic manipulations, it increases the demand for carbon and reduces the feedback inhibition of sucrose. This increases the leaf photosynthesis rate and elevates the total sugar content (*Wu & Birch, 2007*). Manipulation of the genes regulating sucrose isomerase (SI) and ADP-glucose pyrophosphorylase (AGPase) promotes the conversion of sucrose to its isomers and starch. It also increases the sugar content of transgenic tomatoes (*Petreikov et al., 2009*). Some transcription factors are also involved in regulating the metabolism of fruit sugars. For example, the

transcription factor RIN, which is related to fruit maturation, can directly bind to the promoter regions of the vacuolar invertase (*SlVI*) gene and vacuolar invertase inhibitor (*SlVIF*) gene to regulate their expression. This affects sugar metabolism and maturation (*Qin et al., 2016*). ABA (abscisic acid) is essential for the accumulation of sugar in tomato fruits (*Li et al., 2019*). It induces the expression of the NAC transcription factor gene *SlNAP2*. This in turn can regulate ABA synthesis and influence tomato senescence further influencing sugar metabolism. Transgenic tomatoes with RNA interference exhibited a delayed senescence of leaves, increased yield, and higher content of soluble sugar in fruits (*Ma et al., 2018*). The overexpression of *SlGLK* turns the fruit color dark green and significantly increases the starch, sugar, and TSS content. This suggests that there may be a relationship between increased fruit chlorophyll content and enhanced photosynthetic efficiency (*Powell et al., 2012*).

The gene *INVINH1* is an inhibitor of the acid invertase gene *Lin5*, which specifically inhibits cell wall invertase activity. Studies have reported that silencing the invertase inhibitor corresponding to the *Lin5* gene can increase invertase activity. This also significantly increases soluble sugar and TSS content in the fruit (*Jin, Ni & Ruan, 2009*; *Ruan, Jin & Huang, 2009*). The gene *SlVPE5* belongs to the VPE (vacuolar processing enzyme) family of genes. In this family, 15 VPE encoding genes are present in the tomato genome (*Matarasso, Schuster & Avni, 2005*; *Aoki et al., 2010*; *Ariizumi et al., 2011*; *Wang et al., 2016*, *2017*; *Teper-Bamnolker et al., 2017*). *SlVPE5* negatively regulates sugar accumulation (*Ariizumi et al., 2011*; *Wang et al., 2016*). However, the mechanism by which *INVINH1* and *SlVPE5* negatively regulate sugar accumulation has not been determined, and the interactive relationship between *INVINH1* and *SlVPE5* has not been studied.

The clustered regularly interspaced short palindromic repeat (CRISPR)-associated protein 9 (Cas9) genome editing system is a powerful tool for targeted gene modifications in plants. The first publication of the five CRISPR/Cas9 gene editing literatures in plants (*Feng et al., 2013*; *Li et al., 2013*; *Nekrasov et al., 2013*; *Shan et al., 2013*; *Xie & Yang, 2013*) led to explosive growth in this field and enabled the creation of genetic variations. Variations had previously been made randomly by physical radiation and chemical mutagens. However, this method also introduced precise genomic modifications such as deletion, insertion, inversion, and base edits of specific DNA positions in plant genomes. Scientists and plant breeders have the potential to revolutionize crop improvement by developing new plant breeding techniques (NPBTs) (*Shao, Punt & Wesseler, 2018*; *Shan-e-Ali Zaidi et al., 2019*) and usher in a new era of precise and efficient crop improvement.

We analyzed the structural types and genetic relationship of two genes (*INVINH1* and *SlVPE5*) that could inhibit the accumulation of soluble sugar with the goal of increasing soluble sugar content in tomato fruits. Using CRISPR/Cas9 technology, both genes were knocked out separately to obtain mutants (the mutants were named CRISPR-*invinh1* or CRISPR-*vpe5*) with loss of function in each gene. We then analyzed the influence of each gene on glucose, fructose, and TSS during fruit ripening. Through hybridization and self-pollination, we obtained lines with loss of function in both genes. We further explained the regulatory relationship between these genes and the accumulation of soluble sugar and provided significant theoretical guidance for improving

tomato fruit sweetness and trait processing. Our research will aid in the accelerated introgression of a single gene, introduce some elite genes into the breeding lines, and provide a high-efficiency and promising strategy for plant breeding.

## MATERIALS & METHODS

### Plant material and growth conditions

We use cultivated tomato M82 as the experimental material. The tomatoes were grown in a greenhouse at the seedling stage (environmental conditions: 14 h light/10 h dark, daytime temperature of 25 °C, night temperature of 18 °C, and daily automatic irrigation by sprinkler equipment). The seedlings were transplanted to the field before entering the flowering stage. The management of the test field was the same as that of the general field with drip irrigation once a week, nitrogen as the main fertilizer at the seedling stage, and P and K as the main fertilizer at the reproductive stage. The test field was located at the comprehensive testing field (87°47′E, 43°95′N) of Xinjiang Academy of Agricultural Sciences in Urumqi, China.

### Sample preparation

The genotypes of the transgenic seedlings and wild-type seedlings were identified when the plants grew stronger, with at least four leaves, and genomic DNA from the leaves was extracted for the detection of mutations. We tested for soluble sugar and TSS content in the fruit when the fruit was red and ripe (about 50 days post anthesis). At least six uniform fruits (two from each replicate with three replications) were collected from the second and third spikes of each plant, crushed, and homogenized into a sauce. The sauce was then analyzed for soluble sugar and TSS content.

### Phylogenetic analysis

The protein sequences of INVINH and VPE from tomato (*Solanum lycopersicum*), *Arabidopsis thaliana*, pepper (*Capsicum annuum*), citrus (*Citrus sinensis*), soybean (*Glycine max*), tobacco (*Nicotiana tabacum*), rice (*Oryza sativa*), potato (*Solanum tuberosum*), grape (*Vitis vinifera*), and maize (*Zea mays*) were collected using the BLAST database search tool at NCBI (Fig. 1). The following protein sequences were downloaded from GenBank or SGN: Sl INVINH1, accession number AJ010943; Os INVINH1, AK288558; Os INVINH6, AK110798; Zm CWINVINH1, XP_008668976; Zm INVINH2, AX214357; ZM INVINH3, AX214336; At INVINH, AT1G48010; NT VINVINH, AY594179; NT CWINVINH, Y12805; St INVINH1, JQ043180; St INVINH2, GU980594; St VPE, XP_006355320; St VPE2, ACC68681; Nt VPE1b, BAC54828; NtVPE1a, BAC54827; NtVPE3, BAC54830; Nt VPE2, BAC54829; Gm VPE2, NP_001236564; Cs VPE, NP_001275777; Zm VPE, ACG34144; Zm VPE1, AAL58571; Zm VPE2, ACG47915; At VPE, NP_195020; At VPE2, NP_180165; At VPE3, NP_176458; At VPE4, NP_001189938; Vv VPE, AGC54786; Os VPE1, XP_015636414; Ca VPE1b, CA12g18500; Sl VPE1b, Solyc08g065610; Sl VPE5, Solyc12g095910; Sl VPE4, Solyc08g079160; Sl VPE1, Solyc08g065790; Sl VPE2, Solyc08g065780; Sl VPE10, Solyc08g065690; Sl VPE6, Solyc08g065530; Sl VPE11, Solyc08g065710; Sl VPE12, Solyc08g065720; Sl VPE8,

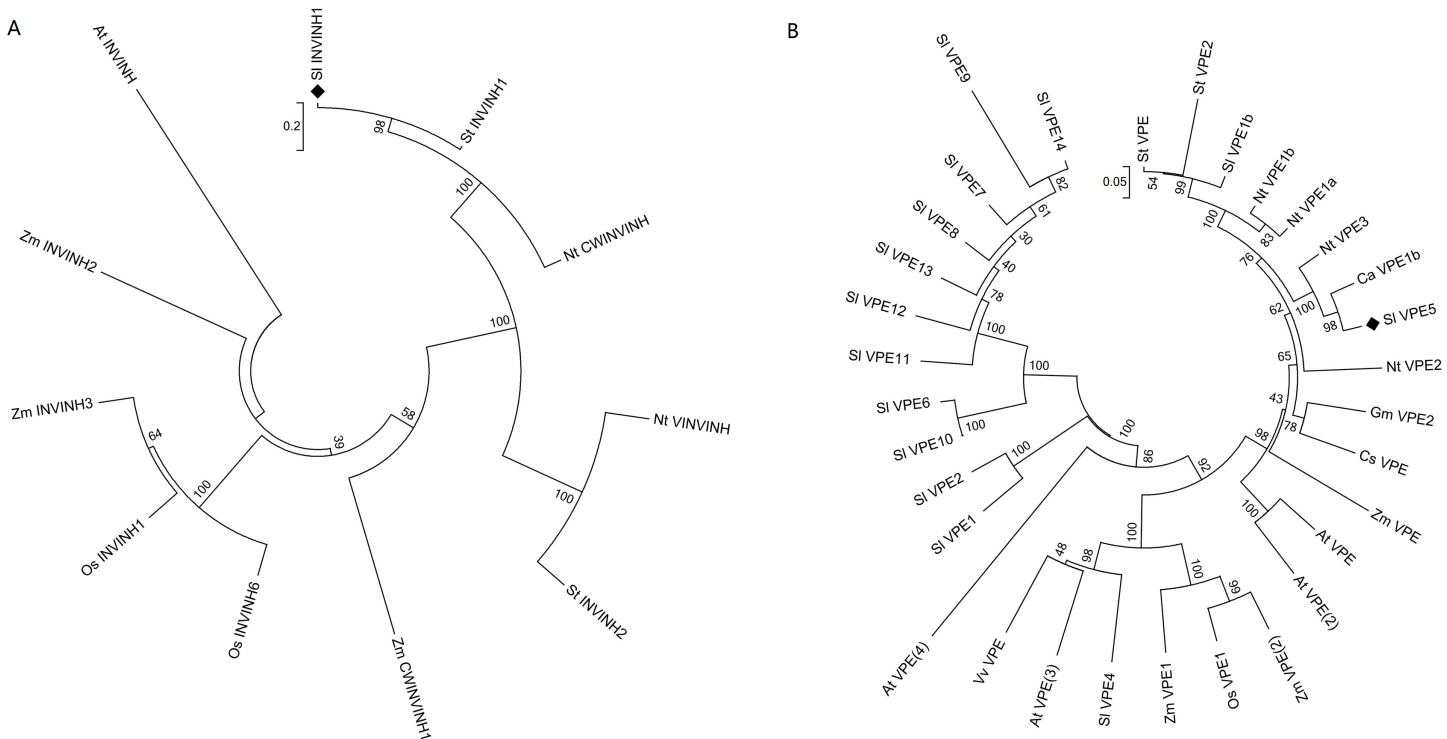

**Figure 1 Phylogenetic analysis of INVINH and VPE proteins in plants.** (A) Phylogenetic tree of INVINH. (B) Phylogenetic tree of VPE. At, *Arabidopsis thaliana*; Ca, *Capsicum annuum*; Cs, *Citrus sinensis*; Gm, *Glycine max*; Nt, *Nicotiana tabacum*; Os, *Oryza sativa*; Sl, *Solanum lyco-persicum*; St, *Solanum tuberosum*; Vv, *Vitis vinifera*; Zm, *Zea mays*. The size bar shows the estimated evolutionary distance.

Solyc08g065590; Sl VPE13, Solyc08g065740; Sl VPE7, Solyc08g065570; Sl VPE9, Solyc08g065600; Sl VPE14, Solyc08g065750. Multiple alignments of the INVINH and VPE protein sequences were processed using MEGA7 (*Kumar, Stecher & Tamura, 2016*). A phylogenetic tree was constructed with the neighbor-joining statistical method using 1,000 bootstrap replicates (*Kumar, Stecher & Tamura, 2016*).

## Design of sgRNA and construction of CRISPR/Cas9 vector

CRISPR-P (http://cbi.hzau.edu.cn/cgi-bin/CRISPR) was utilized to design the target sites of the target genes *SlINVINH1* (Solyc12g099200.1) and *SlVPE5* (Solyc12g095910.1). Each gene was provided with two target sites for a more efficient gene knockout. *SlINVINH1* target sites were selected in the first exon, and *SlVPE5* target sites were selected in the first and second exon (Fig. 2B). The CRISPR/Cas9 original vector was obtained from Prof. Cai-xia Gao (Chinese Academy of Science, Beijing, China). The optimization of CRISPR/Cas9 gene codon and the synthesis of sgRNA sequence was provided by the Biological Company of Hong Xun, Suzhou, China. We used the construction method described by *Yu et al. (2017)*, adding the 35S promoter driving CRISPR/Cas9 gene, the U6 promoter of *Aribidopsis thaliana*, and tomato control the sgRNA1 and sgRNA2, separately. The cloned fragments were assembled using the Circular Polymerase Extension

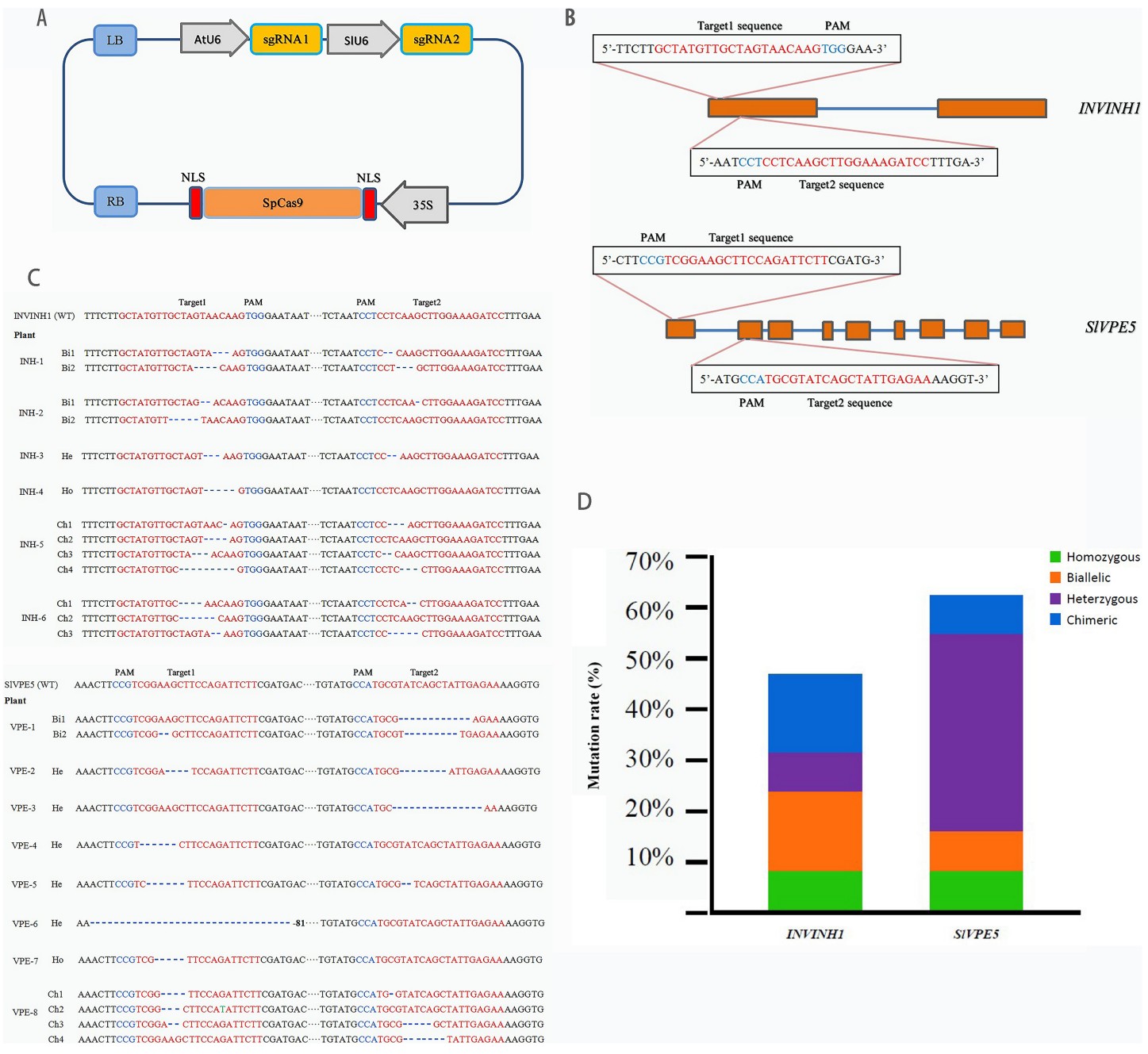

**Figure 2 Target genes editing by CRISPR/Cas9.** (A) The schematic description of CRISPR/Cas9-sgRNA expression cassette. SpCas9 is controlled by a CaMV 35S promoter, the first sgRNA is expressed by the AtU6 promoter, the second sgRNA is expressed by the SlU6 promoter. (B) Four target sites were designed in two target genes. Straight blue lines and rectangular orange boxes are the introns and exons of the target genes, respectively. (C) The sequenced targeted InDel mutation of two target gene. A total of 26 clones of the each PCR amplicon were picked and sequenced. "WT" means Wild-type, "Bi" means Biallele, "He" means Heterozygote, "Ho" means Homozygote, "Ch" means Chimera. (D) Specific types of each target gene in $T_0$ lines. Green, orange, purple, and blue represent homozygous, biallelic, heterozygous, and chimeric mutations, respectively.

Cloning (CPEC) method (*Quan & Tian, 2009*). The pCAMBIA1301 binary vector (AtU6-sgRNA1-SlU6-sgRNA2-35S-Cas9) was constructed and used for gene knockout (Fig. 2A).

## Plant transformation

We used the Agrobacterium-mediated transformation method as described previously (*Van Eck, Kirk & Walmsley, 2006*) to transform pCAMBIA1301 vectors containing the CRISPR/Cas9 and sgRNA cassette into M82. Tomato seeds were germinated after sterilization with 10% NaClO on ½ MS medium. The apical segments of hypocotyls were punctured with Agrobacterium suspension after 9 to 12 days of culturing (OD600 = 0.5–0.6). The explants were then inoculated on a selective medium containing three mg/L hygromycin until the transgenic plants were regenerated from the callus. After the plants were grown *in vitro*, they were transplanted in a pot with soil and placed in light growth chamber.

## DNA extraction and mutation detection

Genomic DNA from fresh frozen leaves was extracted using a high-efficiency plant genome DNA extraction kit (Tiangen, Beijing, China). Specific primers were used to amplify the flanks of genomes containing the target site (Table S1) and the PCR products were isolated by 1% agarose. The PCR products were linked to the pZERO-T vector after cutting and purifying (Transgen, Beijing, China), and 26 positive clones were randomly selected for mutagenesis from each plant. The Sanger method was used for sequencing with M13 primers. We obtains T1 and T2 lines for genotyping analysis by strict self-pollination of the $T_0$ plants. The target fragments were directly sequenced. The genotyping of $F_2$ plants was obtained from the $F_1$ lines using strict self-pollination. The $F_1$ lines were obtained from the two cross-pollinated $T_0$ lines. Their target fragments were also directly sequenced.

## Determinations of soluble sugars, TSS and lycopene

Soluble sugars were determined as described previously by *Freschi et al. (2010)*. Glucose, fructose, and sucrose levels were determined using high-performance anion exchange chromatography with pulsed amperometric detection (HPAEC-PAD; Dionex, Sunnyvale, CA, USA). We used a Carbopac PA1 column (250 × 4 mm, five μm particle size, Dionex, Sunnyvale, CA, USA) as an isocratic run with 18 mM NaOH as the mobile phase. A standard curve made with pure glucose, fructose, and sucrose was used to calculate sugar content. TSS were measured with a refractometer DR201-95 (Kruess, Germany).

For determination of lycopene, pigments were extracted from two g of fresh ripe fruits using acetone/petroleum ether (1:1, v/v), then dried under a stream of $N_2$ and dissolved in 100% dichloromethane. The HPLC analysis was performed with 10 μL dichloromethane-dissolved pigments on ACQUITY UPLC (Waters, Milford, MA, USA). Lycopene was identified by the characteristic absorption spectra (472 nm) and distinctive retention times, compared with the standard substance (Sigma, St. Louis, MO, USA). Each lycopene content was calculated through the linear regression equation generated from the corresponding calibration curve, which was made using the corresponding standard substance.

Each sample contained three replicates with two fruits per replicate.

## RESULTS

### Characteristics and phylogenetic analysis of genes

The invertase inhibitor (INVINH1) protein consists of 171 amino acid residues, with a signal peptide of 19 amino acid residues at the N-terminus. The differences between INVINH1 and another type of invertase inhibitor in tomato, SolyCIF, include distinctions in protein sequences and protein structures (*Reca et al., 2008*). Compared with the sequence of the INVINH1 protein in other crops, four cysteine residues were conserved, which was a significant feature of all invertase inhibitors (*Rausch & Greiner, 2004*). Cluster analysis revealed that the INVINH1 protein of tomato had the closest relationship with that of potato and tobacco, which are also solanaceous crops, but was distantly related to *Arabidopsis*, corn, and rice (Fig. 1A).

The vacuolar processing enzyme (VPE) has been classified in the cysteine protease family and is mainly involved in the regulation of post-translational levels of proteins in vacuoles. The VPE proteins are highly conserved in most plants and animals (*Hara-Nishimura, Takeuchi & Nishimura, 1993*). Among the numerous VPE proteins in tomatoes, SlVPE3 and SlVPE5 have been reported to regulate the accumulation of soluble sugars. The soluble sugar content was higher in RNA interference lines (RNAi-*SlVPE3* and RNAi-*SlVPE5*) compared with the control plants. However, the soluble sugar content of RNAi-SlVPE5 line was significantly higher than that of RNAi-SlVPE3 line. Therefore, only *SlVPE5* was chosen as the target gene for genome editing in this study (*Ariizumi et al., 2011*). Cluster analysis revealed that SlVPE3 and SlVPE5, which are involved in sugar metabolism, did not cluster with other tomato VPE proteases but were rather the most closely related to tobacco (*Nicotiana tabacum*), the NtVPE3 protein, and the sweet pepper CaVPE1b protein (Fig. 1B). This suggests that there may be a class of proteases among the VPE family proteins that specifically regulate the accumulation of sugars.

### Characterization of targeted editing in transgenic plants

The transformation system of the *SlINVINH1* gene knock-out finally yielded 14 $T_0$ generation lines positive for the Cas9 gene. Genomic DNA was extracted from the leaves of each line. PCR using primers designed for the target region was then conducted to amplify the mutant region fragments, and each clone of the amplified products was sequenced and analyzed to obtain information on mutations in the target region of the *INVINH1* gene. The sequencing results demonstrated that among the 14 lines that were positive for Cas9, only six had a mutation in the target sequence of the *SlINVINH1* gene (probability of mutation = 42.86%). Among the mutants, one line was a homozygous mutant (both alleles had the same type of mutation), two were biallelic mutants (both alleles were mutated, but with different mutations), one was a heterozygous mutant (only one allele was mutated), and two were chimeric mutants (only some of the cells are mutated) (Figs. 2C and 2D; Table S2). Detailed mutations in the target sequence were mainly the deletions of single or multiple nucleotides (Fig. 2C).

The transformation system of the *SlVPE5* gene knock-out yielded 13 $T_0$ generation lines that were positive for the Cas9 gene. The results of mutation detection revealed that among the 13 Cas9-positive lines, eight lines had mutations in the target sequence of the *SlVPE5* gene (probability of mutation = 61.54%). Among the mutants, one line was a homozygous mutant, another line was a biallelic mutant, five lines were heterozygous mutants, and one line was a chimeric mutant (Figs. 2C and 2D; Table S2). Detailed mutations in the target region were mainly the deletion and replacement of single or multiple nucleotides (Fig. 2C). Further analysis revealed that the mutation types of the *SlINVINH1* gene were mainly biallelic mutations and chimeras, while the mutation type of the *SlVPE5* gene was mainly heterozygotic (Figs. 2C and 2D; Table S2).

## CRISPR/Cas9-mediated mutants exhibited soluble sugar and TSS content increase

The $T_0$ mutant line was obtained as described above. Soluble sugar and TSS contents were determined after maturation with three replicates for each line. Among the $T_0$ mutant lines of the *SlINVINH1* gene four lines (INH-1, INH-2, INH-3, and INH-4) (Fig. 2C) were selected. The INH-5 and INH-6 lines were not selected as they were chimeras and displayed an extremely low probability of passing the mutations on to the next generation. We observed that the contents of both glucose and fructose in the fruits from lines INH-1, INH-2, and INH-4 were significantly increased compared to the wild-type. INH-1 showed a 40.19% increase in glucose and a 42.42% increase in fructose; INH-2 showed a 36.39% increase in glucose and a 35.69% increase in fructose; INH-4 showed a 40.82% increase in glucose and a 42.76% increase in fructose (Fig. 3A; Table S3). In addition, the TSS content also showed a significant increase (31.90% in INH-1, 30.17% in INH-2, 32.76% in INH-4) (Fig. 3A; Table S3), indicating that the *SlINVINH1* gene mainly regulated the accumulation of glucose and fructose in the fruits. However, the content of soluble sugars in the fruit of heterozygous INH-3 did not increase significantly compared to the wild-type (Fig. 3A; Table S3), which could be due to the recessive function of the mutated *slinvinh1* gene.

Among the $T_0$ lines that carried the mutations of the *SlVPE5* gene, three lines, VPE-1, VPE-2, and VPE-7, were selected. The VPE-8 line was not selected as it was chimera and showed an extremely low probability of passing the mutations to the next generation. Among the five heterozygous, only one line, namely VPE-2, was selected as a representative for testing. The test revealed that the glucose and fructose content in the lines fruits of VPE-1 and VPE-7 increased significantly (glucose increased 35.20% in VPE-1, and increased 35.83% in VPE-7; fructose increased 37.33% in VPE-1, and increased 43.00% in VPE-7). The TSS content also increased significantly (30.63% in VPE-1, 32.43% in VPE-7) (Fig. 3B; Table S3), indicating that the *SlVPE5* gene also regulated the accumulation of glucose and fructose in fruits. However, the soluble sugar content in the fruits of heterozygous INH-2 did not increase significantly compared to the wild-type (Fig. 3B; Table S3), which might also be caused by the recessive gene characteristics of the mutant *slvpe5* gene.

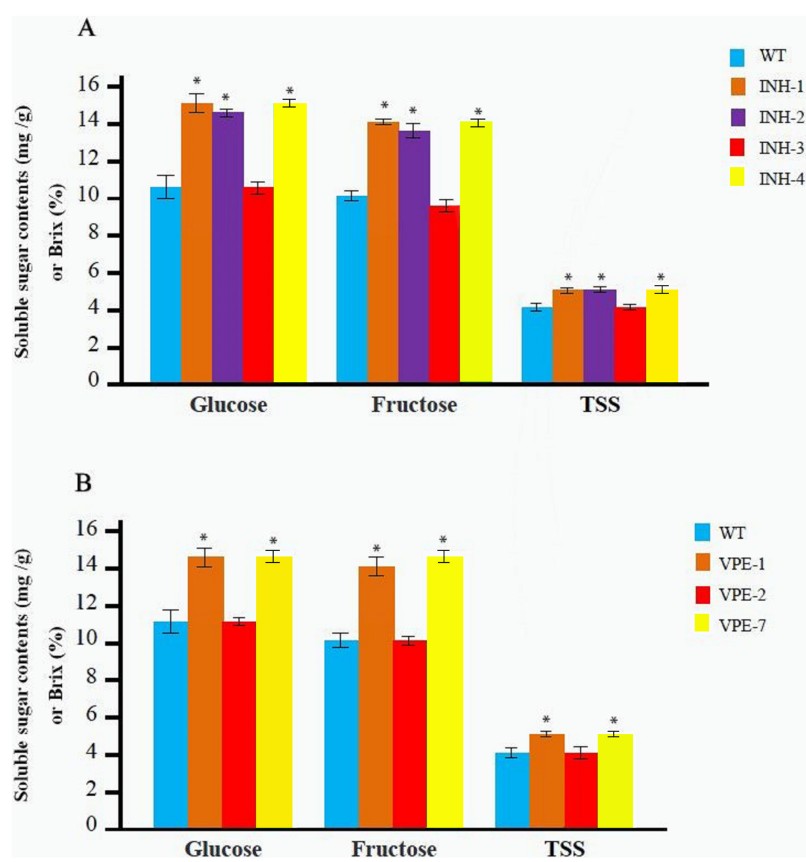

**Figure 3 Determination of the soluble sugar and TSS content in red fruit from different $T_0$ lines and WT.** (A) The contents of the soluble sugar and TSS of red fruit in four different $T_0$ lines of editing *SlINVINH1* gene and WT. (B) The contents of the soluble sugar and TSS of red fruit in three different $T_0$ lines of editing *SlVPE5* gene and WT. The contents of glucose and fructose is measured in mg/g in fruit weigh, and the TSS is measured in Brix (%). *($P < 0.05$, Student's *t*-test, $n = 3$) indicate statistically significant differences between $T_0$ mutant lines and wild-type.

## The double homozygous mutants further increased the soluble sugar and TSS content

The $T_0$ generation homozygous mutant lines INH-4 and VPE-7 were cross-pollinated to generate the $F_1$ generation. The $F_2$ generation was obtained by self-pollination of the $F_1$ generation, and 515 well-grown lines were obtained after sowing the $F_2$ generation seeds. We screened for the gene segregation and obtained 32 lines with homozygous mutated *SlINVINH1* and *SlVPE5* sites (genotype: *inh/inh-vpe/vpe*) from 515 lines. We identified two lines without the exogenous Cas9 genes by further screening for the exogenous Cas9 genes among the 32 lines. The two lines were numbered $F_2$–1 and $F_2$–2.

The soluble sugar and TSS in fruit from the $F_2$–1 and $F_2$–2 lines were determined after maturation, each line with three replicates. Glucose and fructose content in $F_2$–1 and $F_2$–2 fruits significantly increased compared to the wild-type. In $F_2$–1, glucose increased by 64.86%, and fructose increased by 68.40%, while in $F_2$–2, glucose increased by

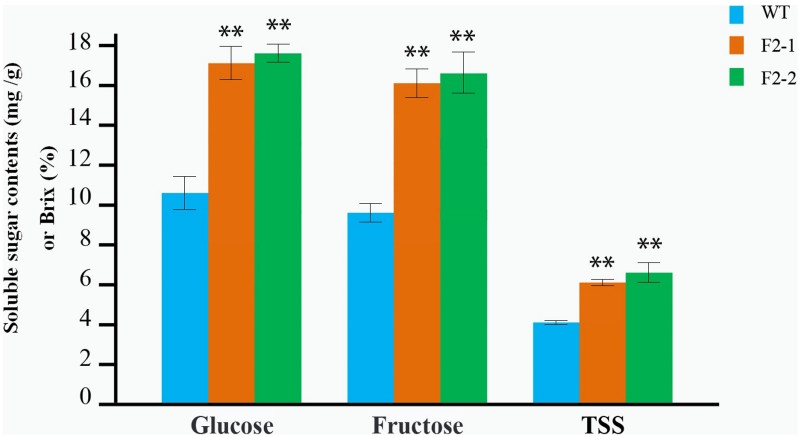

**Figure 4 Determination of the soluble sugar and TSS content in red fruit from the WT and the two F₂ lines of double homozygous mutants of edited *SlINVINH1* and *SlVPE5* genes.** The contents of glucose and fructose is measured in mg/g in fruit weigh, and the TSS is measured in Brix (%). **($P < 0.01$, Student's *t*-test, $n = 3$) indicate statistically highly significant differences between F₂ lines and wild-type.

67.41%, and fructose increased by 69.44% (Fig. 4; Table S4). The TSS content also showed a significant increase (55.17% in F₂–1 and 62.07% in F₂–2) (Fig. 4; Table S4).

We observed that both the *SlINVINH1* and *SlVPE5* gene mainly regulated the accumulation of glucose and fructose in the fruit. In addition, both glucose and fructose content was significantly synergistically increased in the F₂–1 and F₂–2 fruit (compared to the average of INH-4 and VPE-7 lines, glucose increased by an average of 18.12% in two F₂ lines; and fructose increased by 14.06%) (Table S5). The TSS content was also significantly higher than the single locus-mutated homozygote (which increased by 22.31%) (Table S5). This indicated that the *SlINVINH1* and *SlVPE5* genes displayed a synergistic effect in regulating the content of soluble sugar.

## Dynamic variation patterns of fruit development and coloration

*SlINVINH1* is an invertase inhibitor gene and the loss of invertase function may delay fruit ripening. However, the loss of *SlVPE5* gene function may also cause changes in other commercial traits like the size and color of fruits. We followed the development of two lines (F₂–1 and F₂–2), starting from flowering to fruit ripening, in real time. The results indicated that tomatoes of the F₂–1 and F₂–2 lines were similar to the wild-type tomato fruits, starting from flowering to the physiological fruit expansion stage, and including both shape and size of the fruits. However, during the breaker stage and the beginning of the ripping stage, fruits of the F₂–1 and F₂–2 lines were significantly delayed the break stage compared to the wild-type fruits, although they were fully ripened at the same time as wild-type fruits (Fig. 5A). This indicated that although the knock-out of the *SlINVINH1* and *SlVPE5* genes did delay the break stage and the color change of tomato fruits, it neither affected the ripening and harvesting time of the fruits nor the color of fruits after ripening nor did it affect fruit weight (Fig. 5B; Table 1).

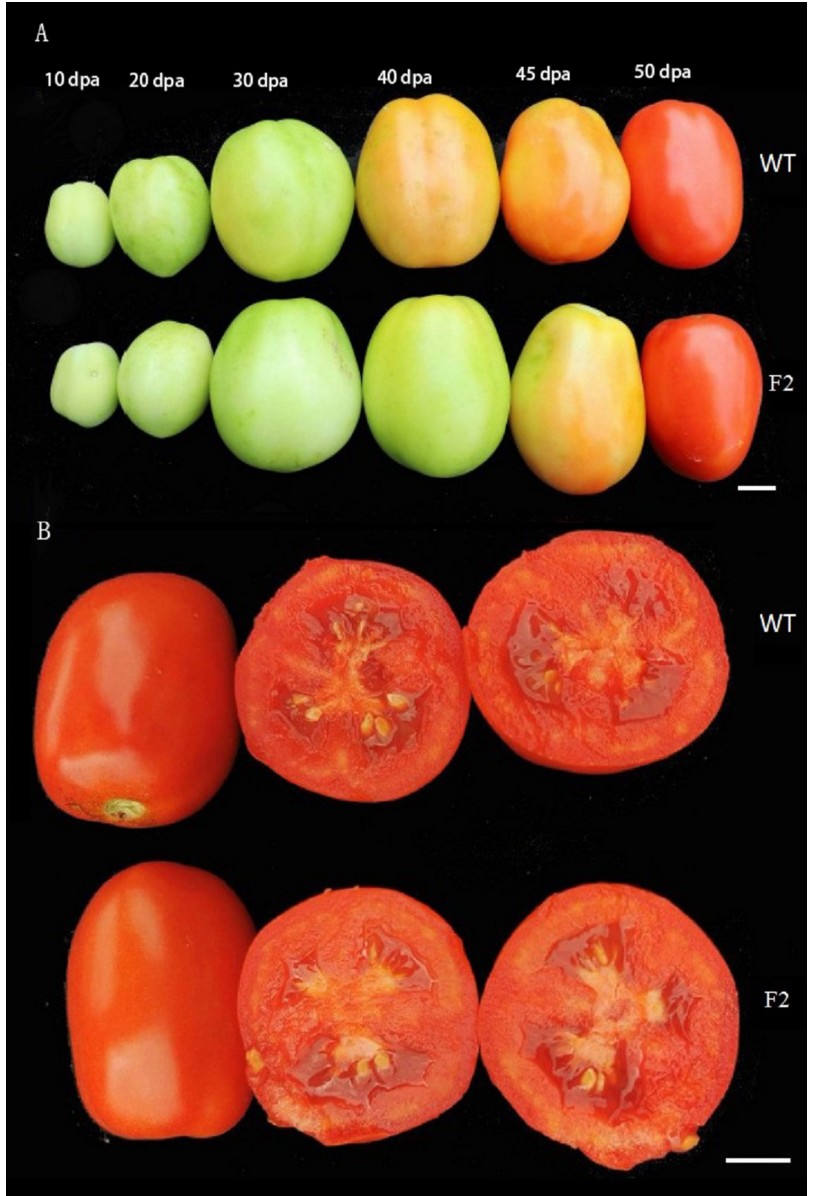

**Figure 5 Phenotypic detection of the WT fruits and the two F$_2$ lines fruits.** (A) Developmental series of WT fruits (up) and F$_2$ fruits (down), "dpa" is days post anthesis. (B) Comparison of bisected fruit at the ripening stage. Bars = one cm.

## DISCUSSION

### The effects of SlINVINH1 and SlVPE5 in increasing the soluble sugar and the TSS content of tomato

*SlINVINH1* is an invertase inhibitor gene, which specifically inhibits the activity of cell wall invertase. Invertase hydrolyzes sucrose into glucose and fructose and plays a major role in fruit development. The suppression of *SlINVINH1* gene expression increases the glucose and fructose content in fruits (*Sturm, 1999*; *Jin, Ni & Ruan, 2009*). *SlVPE5* belongs to the family of VPE genes, which targets various types of vacuolar hydrolases

**Table 1 Comparison of lycopene content and weight of red fruit between WT and $F_2$ lines.**

| Lines | Lycopene mg/100 g fw | | | Weight g | | |
|---|---|---|---|---|---|---|
| | Rep | Avg | *P*-value | Rep | Avg | *P*-value |
| WT | 12.8 | 13.37 ± 0.54 | | 79.8 | 80.40 ± 2.24 | |
| | 13.2 | | | 83.4 | | |
| | 14.1 | | | 78.0 | | |
| F2-1 | 13.0 | 13.07 ± 0.33 | 0.741 | 80.2 | 78.73 ± 1.04 | 0.751 |
| | 13.5 | | | 77.9 | | |
| | 12.7 | | | 78.1 | | |
| F2-2 | 14.2 | 13.63 ± 0.45 | 0.786 | 83.8 | 81.03 ± 2.98 | 0.956 |
| | 13.1 | | | 76.9 | | |
| | 13.6 | | | 82.4 | | |

Note:
WT, Wild-type; fw, fruit weight; Rep, Three test replicates; Avg, Average.

(such as β-glycosidase, α-mannnosidases, and α-galactosidases) for degradation and the negative regulatory mechanism for soluble sugar accumulation (*Rojo et al., 2003*; *Ariizumi et al., 2011*). We observed that the both genes *SlINVINH1* and *SlVPE5* mainly regulated the accumulation of glucose and fructose in fruits (Figs. 3A, 3B, and 4), confirming the findings of previous research (*Jin, Ni & Ruan, 2009*; *Ariizumi et al., 2011*; *Xu et al., 2017*). Previous research also found that the *SlVPE5* gene also regulates the accumulation of sucrose in fruits (*Wang et al., 2016*). However, in this study no sucrose content was detected in mature fruits (detection threshold was ≥0.2 mg/g fw) in the $T_0$ or $F_2$ lines. This may be due to the low sucrose content in mature fruits and because the sucrose contribution to the TSS of mature fruits is very small (*Tieman et al., 2017*). The knock-out of both loci in homozygotes resulted in a significant increase in both tested sugars in the fruit, and the TSS content was also significantly higher than that in single locus-mutated homozygotes, indicating that there was a synergistic effect of *SlINVINH1* and *SlVPE5* genes in regulating the content of soluble sugar. This proves once again that the increase of soluble sugar content in fruits is a complex process and not regulated by a single gene (*Beckles et al., 2012*; *Patrick, Botha & Birch, 2013*). This finding is crucial for the improvement of quantitative genetic traits of soluble sugar.

## The knockout efficiency of two genes

The knock-out efficiency of both genes were different. In the knock-out experiment on *SlINVINH1*, we obtained six lines with the successful knock-out of the target gene from 14 positive lines. Its knock-out efficiency was 42.86%. In the knock-out experiment on *SlVPE5*, eight lines with the successful knock-out of the target gene were obtained from 13 positive lines, with a knock-out efficiency of 61.54%. Furthermore, the mutation types of the *SlINVINH1* gene were mainly biallelic mutations and chimeras, while the mutation types of the *SlVPE5* gene were mainly heterozygous (Figs. 2C and 2D; Table S2). This may be linked to the different types and structures of the genes (*Pan et al., 2016*; *Li et al., 2018*). According to previous research vector compontents, such as codon

optimization of CRISPR/Cas9 gene, promoter, and structure of gRNA, may affect gene knockout efficiency (*Dang et al., 2015*; *Ma et al., 2015*; *Čermák et al., 2017*; *Feike et al., 2019*; *Shao et al., 2020*; *Grützner et al., 2021*).

### Effective improvements to the commercial traits of tomato

We obtained two homozygous lines (F$_2$–1 and F$_2$–2) with mutations in both the functional gene loci, excluding exogenous Cas9, and found that both the soluble sugar and TSS content in tomato fruits were significantly increased (Fig. 4), facilitating the rapid improvement of the commercial traits of tomatoes. Our study confirmed the feasibility of target gene editing by the transfer of Cas9 and sgRNA, and the subsequent application of conventional methods such as hybridization and inbreeding. Two mutation sites may be brought together, however, the mutation site could be made homozygous and the exogenous gene Cas9 could be eradicated. This method greatly shortened the breeding cycle to improve tomato varieties. It also completely eliminated various risk factors associated with the exogenous genes. This system will improve technological innovation and progress in tomato breeding, and has tremendous potential in a variety of applications.

## CONCLUSIONS

The purpose of this study was to further elucidate two tomato genes (*SlINVINH1* and *SlVPE5*) that inhibited the accumulation of soluble sugars in tomato fruits. We also aimed to obtain the knock-out lines of two genes using CRISPR/Cas9. We determined the aggregated lines with both CRISPR-*invinh1* and CRISPR-*vpe5* using hybridization and self-pollination. The glucose, fructose, and TSS content of CRISPR-*invinh1* and CRISPR-*vpe5* were significantly increased compared to wild-type tomato lines. In addition, the levels of glucose, fructose, and TSS further improved in the lines with CRISPR-*invinh1* and CRISPR-*vpe5* simultaneously *versus* that of the single gene knock-out lines. This result indicated that the two genes had a synergistic effect in increasing the content of these soluble sugars. Thus, the knock-outs, *SlINVINH1* and *SlVPE5*, may effectively increase the content of soluble sugars in tomato and provide an important theoretical and practical basis for improving the sweetness of tomato fruits and their processing quality.

### Funding

The work was funded by the National Natural Science Foundation of China (31860555, 31760581), the Special Incubation Project of Science & Technology Renovation of Xinjiang Academy of Agricultural Sciences (xjkcpy-2021001), and the National Key R&D Program of China (2017YFD0101906). The funders had no role in study design, data collection and analysis, decision to publish, or preparation of the manuscript.

### Grant Disclosures

The following grant information was disclosed by the authors:
National Natural Science Foundation of China: 31860555, 31760581.

Special Incubation Project of Science & Technology Renovation of Xinjiang Academy of Agricultural Sciences: xjkcpy-2021001.
National Key R&D Program of China: 2017YFD0101906.

## Competing Interests

The authors declare that they have no competing interests.

## Author Contributions

- Baike Wang conceived and designed the experiments, performed the experiments, analyzed the data, prepared figures and/or tables, authored or reviewed drafts of the paper, and approved the final draft.
- Ning Li performed the experiments, analyzed the data, prepared figures and/or tables, authored or reviewed drafts of the paper, and approved the final draft.
- Shaoyong Huang performed the experiments, prepared figures and/or tables, and approved the final draft.
- Jiahui Hu performed the experiments, prepared figures and/or tables, and approved the final draft.
- Qiang Wang performed the experiments, prepared figures and/or tables, and approved the final draft.
- Yaping Tang performed the experiments, authored or reviewed drafts of the paper, and approved the final draft.
- Tao Yang performed the experiments, authored or reviewed drafts of the paper, and approved the final draft.
- Patiguli Asmutola performed the experiments, authored or reviewed drafts of the paper, and approved the final draft.
- Juan Wang conceived and designed the experiments, prepared figures and/or tables, authored or reviewed drafts of the paper, and approved the final draft.
- Qinghui Yu conceived and designed the experiments, prepared figures and/or tables, authored or reviewed drafts of the paper, and approved the final draft.

## Data Availability

   The raw measurements are available in the Supplemental File.

## Supplemental Information

Supplemental information for this article can be found online at http://dx.doi.org/10.7717/peerj.12478#supplemental-information.

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
