# Peer review of "Enhanced soluble sugar content in tomato fruit using CRISPR/Cas9-mediated SlINVINH1 and SlVPE5 gene editing"

_PeerJ, doi:10.7717/peerj.12478_

## Round 0.1 · original submission · Major Revisions

According to both Reviewers, I recommend a strong English editing work, in order to improve the flow of the text. Therefore, I strongly suggest the revision by a professional English proofreader. Figures and tables also could be much improved.

The “Material and Methods” and Discussion” sections, need to be reinforced and better structured, by describing in detail the different experimental procedures, and reinforcing the critical review of the literature, as well as addressing the significance of the results obtained in comparison with the literature.

Please, modify the manuscript according to the suggestions of the reviewers, and prepare a detailed point-to-point response to the criticisms they raised.

Reviewer 1 ·

Basic reporting

The english needs an improvement. i corrected in the pdf some parts, but I think that should be revised by a native english speaker.
The literature references are appropriate, but I suggest the authors to improve the introduction with a section focused on a brief story on tomato editing approaches, due to the strong application of this technique in this specie.
The reported results are reliable and congruent with the experimental plan. However the discussion about the phenotype development of F2 and wild type, needs to be improved, as marked in the pdf.

Experimental design

The research fits the aims and scope of the journal. The genetic analysis is appropriate, but needs to be improved as reported in the revised pdf (comparison between T0 and F2 values).
Due to numerous application of editing in tomato, the authors should underline, the novelty of this work, that seems to be non so relevant.
The methods are sufficient described, but they could be improved explaining the experimental plan in a detailed way.

Validity of the findings

As previously described, the authors should better underline the novelty of their application, and the validity of the conducted research, respect the others previously conducted.
The data are sufficient, the number of T0 lines is not so high, and the conclusions related to the efficiency of the promoters are not statistic supported.
The final discussion about the phenotypic analysis of F2 and wild type fruits is only qualitative and not metric. Please see the comment in the text

Additional comments

Dear authors, the work needs to be carefully revised and there are some methodological points that are not appropriate developed and supported by quantitative analyses.

Annotated reviews are not available for download in order to protect the identity of reviewers who chose to remain anonymous.

Reviewer 2 ·

Basic reporting

English of this manuscript is understandable. However, the English quality is low. This manuscript should be proofread by a professional English proofreader.

I think that sufficient references, field background and context were not provided in this manuscript. Therefore, readers cannot understand background and importance of this study precisely.

Quality and resolution of Figure 1 and Figure 2 are very low.

Experimental design

Aims and scope of this study are clear. However, the presented data and explanations are not enough to tell importance of this study.

Methods was described too roughly and poorly and do not provide sufficient information to be reproducible by another investigator.

Validity of the findings

No comment.

Additional comments

This manuscript entitled “Enhanced soluble sugar content in tomato fruit by CRISPR/Cas9-mediated two genes editing” by Baike Wang et al. reports the production of knock out tomato of two genes (SlINVINH1 and SlVPE5) by CRISPR/Cas9 and increase of glucose, fructose and total soluble solids (TSS) in it fruits. I found high value from both plant physiological and agricultural points of view in this study. However, presented data is not enough to tell the importance of this study and preparation of manuscript is very rough and poor. Comments and requests are written below.


[Major points]

Explanation of experiments and results and discussion of this manuscript is very rough and poor. They should be revised thoroughly. All authors should check again this manuscript and contribute to improve the manuscript.

The mechanism why knockout of SlINVINH1 and SlVPE5 leaded the increase of glucose, fructose and TSS was not discussed. This should be explained clearly and discussed.

Methods was described too roughly and poorly and do not provide sufficient information to be reproducible by another investigator. Methods should be revised.

L288-301 The section “Dynamic variation pattern of the fruit development and coloring”: This is the most critical point. All phenotypic data, including Figure 5, are not quantitative. Quantitative data should be presented. At least data showing fruit size or weight is necessary.

Quality and resolution of Figure 1 and Figure 2 are very low. The figures should be revised. How many genes encoding INVINH and VPE do present in tomato genome? This should be mention in the text. In addition, gene ID for all genes appeared in Figure 1 should be described.


[Minor points]

L32- “In this study, two genes (SlINVINH1 and SlVPE5) that inhibited the accumulation of soluble sugar in tomato fruits were identified” and L347- “The purpose of this study was to identify two tomato genes (SlINVINH1 and SlVPE5)”. SlINVINH1 and SlVPE5 had already been identified as key genes of sugar accumulation in tomato in the previous studies (Jin, Ni & Ruan, 2009; Ruan, Jin & Huang, 2009; Ariizumi et al., 2011; Wang et al., 2016). Therefore, they are not new finding in this study.

L92: Lin 5 is INVINH1. This is written in L116, but it should be mention first appearance.

L97: “maltulose and kestose” should be change to “fructose and glucose”.

L133: “practical basis for the improvement of the flavor of tomato fruits and trait processing”. I don’t understand why this study relate to the improvement of the flavor of tomato fruits.

L206-: “Among the numerous VPE proteins in tomatoes, SlVPE3 and SlVPE5 have been reported to regulate the accumulation of soluble sugars (Ariizumi et al., 2011)”. Why only SlVPE5 was chosen as the target gene of genome editing in this study?

L220-: I don’t understand the meanings of “homozygous mutant”, “biallelic mutant”, “heterozygous mutant” and “chimeric mutants” clearly. Explanation and definition of them are necessary.

L233-239 “In addition, at target-1 site of the SlINVINH1 gene, there are 14 mutations …… the Arabidopsis U6 promoter was higher than that of the tomato U6 promoter” and L326-333 “The sgRNA initiated by the U6 promoter of Arabidopsis thaliana …. sgRNA may also cause differences in the rate of mutation, which is consistent with the findings of previous studies (Ma et al., 2015; Čermák et al., 2017; Shao et al., 2020)”. These descriptions are too speculative and should be removed.

L289: “Because the SlINVINH1 is an invertase gene, loss of invertase function may delay fruit ripening”. This is wrong description. “SlINVINH1 is an invertase inhibitor gene”.

---

## Round 0.2 · Major Revisions

Although reviewers found that the manuscript was noticeably improved by revision, they also emphasized that additional revisions are needed to make this work acceptable. Please address the remaining issues of the reviewers and amend your manuscript accordingly.

Reviewer 1 ·

Basic reporting

The english sounds professional and clear. The references are correctly provided. The material and methods and the discussion sections are revised and implemented.

Experimental design

The experimental design is more clear and complete. The authors followed the reviewer comments and although the statistical analysis is not so strong, the work is more clear and inserted in the scientific context

Validity of the findings

The revision addressed the problems of the impact and novelty of the work.

Reviewer 2 ·

Basic reporting

English of this manuscript is understandable. I found several unsuitable/not understandable scientific words. They were pointed out in the following comment.

References, field background and context were added suitably to this manuscript.

Resolution of Figure 1 and Figure 2 are still low.

Experimental design

The manuscript was improved and now the scope of this study became clear.

The methods were improved suitably. Method for measurement of lycopene is missing, thus it should be added.

Validity of the findings

No comment.

Additional comments

I was satisfied with the revised manuscript mostly. However I still have requests and comments as described below.

>>Comment 5: Quality and resolution of Figure 1 and Figure 2 are very low. The figures should be revised. How many genes encoding INVINH and VPE do present in tomato genome? This should be mention in the text. In addition, gene ID for all genes appeared in Figure 1 should be described.
>Respond to this comment: We have made improvement of Figure 1 and Figure 2 to make it more legible.
Reply to the response: Resolution of Figure 1 and Figure 2 are still low.

>>Comment 4: L133: “practical basis for the improvement of the flavor of tomato fruits and trait processing”. I don’t understand why this study relate to the improvement of the flavor of tomato fruits.
>Respond to this comment:The flavor of fruit includes smell and taste, and soluble sugar is the most important component of fruit taste (Mathieu et al., 2009; Tieman et al., 2017). This study increased the amount of soluble sugar in the tomato fruit, so we think this study is related to the improvement of the flavor of tomato fruits.
Reply to the response: As you mention, “flavor” includes smell and taste and soluble sugar is the most important component of taste. In this study, smell of fruit was not checked and smell may not be changed in the genome edited tomato. Therefor “flavor” is not suitable word at least in this sentence and also in other sentences in this text. “taste” or “sweetness” is suitable word in this study.

>>Comment 5: L206-: “Among the numerous VPE proteins in tomatoes, SlVPE3 and SlVPE5 have been reported to regulate the accumulation of soluble sugars (Ariizumi et al., 2011)”. Why only SlVPE5 was chosen as the target gene of genome editing in this study?
>Respond to this comment: It is really true as Reviewer comment that the soluble sugar content was higher in RNA interference lines (RNAi-SlVPE3 and RNAi-SlVPE5) compared with the control plants. However, the soluble sugar content of RNAi-SlVPE5 line was significantly higher than that of RNAi-SlVPE3 line (Ariizumi et al., 2011). Therefore, only SlVPE5 was chosen as the target gene for genome editing in this study.
Reply to the response: Add this explanation in the text.


Additional comments

Why don’t the authors include “SlINVINH1 and SlVPE5” in the title?

“CRISPR-invinh1” and “CRISPR-vpe5” were used only in Abstract and Conclusion and they are not defined.

L31: “or activating the key protein kinases” is not clear. It had better to be removed.

L88: “Insoluble acid invertase activity decreases significantly in the cell wall” had better to be changed to “The activity of insoluble acid invertase, also called cell wall invertase, decreases significantly”.

L93: “uptake” had better to be changed to “unload”.

L94: “Lin5 influences the biosynthesis of cuticle components. This indicates that the process is coupled with sugar uploading by a mechanism which links the carbon supply with the capacity for fruit expansion (Vallarino et al., 2017)” is not clear discussion. I suggest to remove these sentences.

Supplementary Table 5 should be moved to the main text with statistical analysis. Table 1 had better to be moved to supplemental data.

L99: The meaning of “pulp” is unclear in this sentence. Use more suitable word.

L118: I sont agree with “The gene SlGLK, which controls the shoulder development of tomato fruits”, because GLK is a master regulator of chlorophyll biosynthesis.

L180: “paper” is wrong. It is “pepper”.

L184-L207: “accession number” other than the first one should be removed.

Quality of the supplementary data is poor. Prepare high quality tables.

---

## Round 0.3 · accepted · Accept

Thank you very much for addressing all the critiques of the reviewers, who were completely satisfied by the revision. The revised manuscript is acceptable now.

Reviewer 2 ·

Basic reporting

English of this manuscript is understandable.

References, field background and context were added suitably to this manuscript.

Figures and Tables are presented suitably.

Experimental design

The manuscript was improved and the scope of this study became clear.

The methods were improved suitably.

Validity of the findings

No comment.

Additional comments

I respect the work to improve this manuscript by the authors. This manuscript had revised according to my comments and requests properly. I have no more comment or request.